# Perspectives for Targeting Ezrin in Cancer Development and Progression

**Jean Carlos Lipreri da Silva, Hugo Passos Vicari and João Agostinho Machado-Neto ***

Department of Pharmacology, Institute of Biomedical Sciences, University of São Paulo,
São Paulo 05508-000, Brazil

* Correspondence: jamachadoneto@usp.br; Tel.: +55-11-3091-7467

**Abstract:** Recent advances have been made in understanding molecular markers involved in cancer malignancy, resulting in better tumor staging and identifying new potential therapeutic targets. Ezrin (EZR), a member of the ezrin, radixin, moesin (ERM) protein family, is essential for linking the actin cytoskeleton to the cell membrane and participates in the signal transduction of key signaling pathways such as Rho GTPases and PI3K/AKT/mTOR. Clinical and preclinical studies in a wide variety of solid and hematological tumors indicate that (i) EZR is highly expressed and predicts an unfavorable clinical outcome, and (ii) EZR inhibition reduces proliferation, migration, and invasion in experimental models. The development of pharmacological inhibitors for EZR (or the signaling mediated by it) has opened a new round of investigation, but studies are still limited. The scope of the present review is to survey studies on the expression and clinical impact of EZR in cancer, as well as studies that perform interventions on the function of this gene/protein in cancer cells, providing proof-of-concept of its antineoplastic potential.

**Keywords:** ezrin; ezrin/radixin/moesin (ERM) protein family; cancer; antineoplastic agent

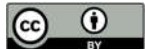

## 1. Introduction

Ezrin (EZR) is a protein member of the ezrin, radixin, moesin (ERM) family, and its function, when phosphorylated, is essential for linking the actin cytoskeleton to the cell membrane. The FERM domain is composed of three structural modules (F1, F2, and F3), which together form a compact clover-shaped structure and bind to integral membrane proteins, adhesion molecules, multidrug resistance proteins, scaffold proteins, Rho-related proteins, and tyrosine kinase proteins, and are involved in epithelial organization, villus morphogenesis, and $PIP_2$ signaling. EZR plays an essential role as an activator of notable signal transduction pathways involved in cancer progression, such as PI3K/AKT/mTOR signaling, which provides a mechanism to anchor PI3K in the proximity of its substrate recruiting the p85 regulatory subunit. In contrast, the phosphorylation of EZR at Y353 is also essential to activate the PI3K signaling [1–5]. The activation of AKT protects cells from apoptosis by phosphorylating and inactivating BAD, a proapoptotic member of the BCL2 family, and increasing cell proliferation [6]. Furthermore, ERM phosphorylation, which appears to be regulated positively by Rho and (possibly) negatively by RAC, may activate downstream signaling of Rho proteins (including RAC) that are required for membrane ruffling and lamellipodium extension and CDC42 that induces the formation of filopodia; both GTPases are essential for cell migration and invasion [7,8] (Figure 1). Thus, EZR may serve as a proliferation- and metastasis-related oncogene by modulating multiple cellular processes [9]. EZR is widely expressed in normal tissues, but its expression is time-specific during human development. For example, EZR immunoreactivity is positive in early embryo stages but undetectable during later development and postnatal stages [10]. Recently, reports have shown that EZR is upregulated in several human tumors, as described in the following.

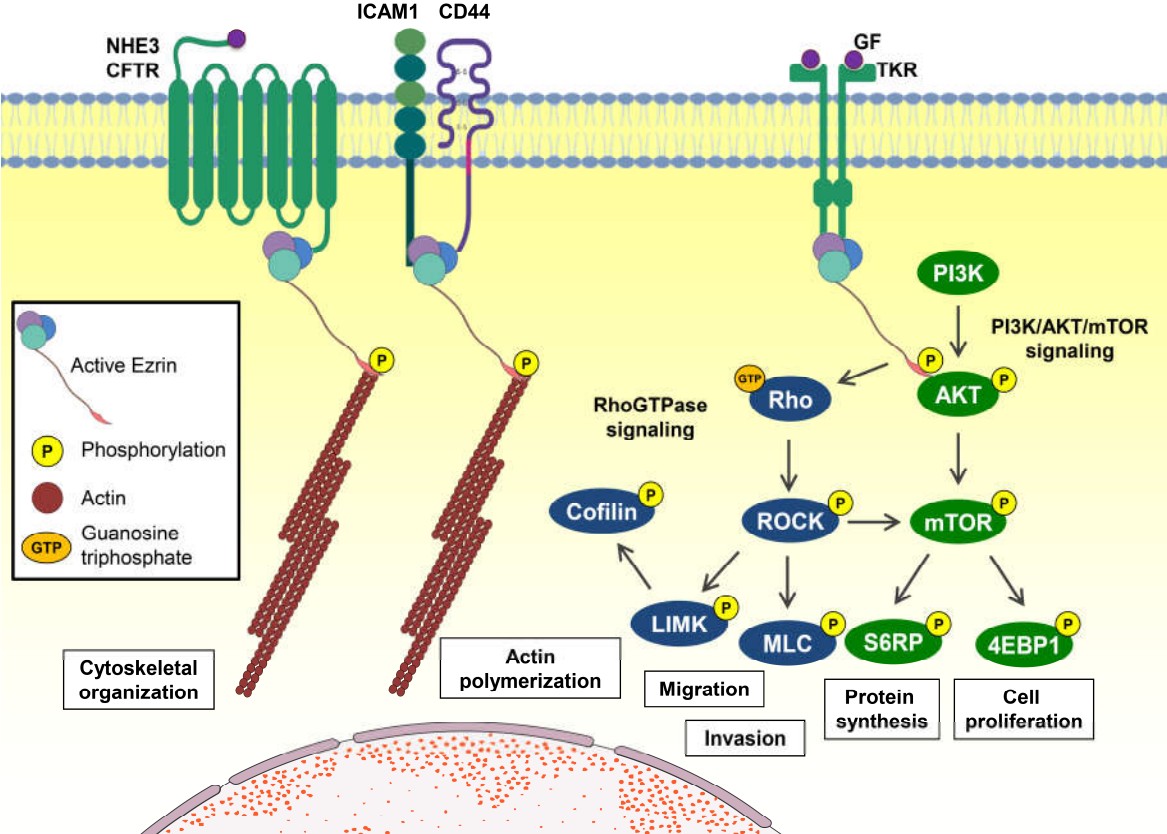

**Figure 1.** Ezrin-mediated signaling pathways and cellular processes. When phosphorylated, ezrin (EZR) may bind membrane proteins such as G protein-coupled receptors (GPCRs) and other receptors allowing the binding of F-actin (actin filaments) with the membrane, promoting reorganization of the cytoskeleton. EZR also binds to several transmembrane receptors, such as growth factor or cytokine receptors, promoting the activation of multiple signaling pathways. Among the main signaling pathways triggered by EZR, we highlight the PI3K/AKT/mTOR pathway, which promotes protein synthesis and cell proliferation, and the Rho GTPase pathway, which promotes migration, invasion, and actin polymerization. Abbreviations: NHE3, Na+/H+ exchanger; CFTR, cystic fibrosis transmembrane conductance regulator; ICAM1, intercellular adhesion molecule 1; CD44, cell surface adhesion receptor; GF, growth factor; TKR, tyrosine kinase receptor; P, phosphorylation; GTP, guanosine triphosphate; PI3K, phosphatidylinositol 3-kinase; AKT, protein kinase B; mTOR, mammalian target of rapamycin; S6RP, ribosomal protein S6; 4EBP1, eukaryotic initiation factor 4E-binding protein 1; Rho, homologous Ras proteins; ROCK, Rho kinase; MLC, myosin light chain; LIMK, LIM kinase.

The EZR C-terminal domain, especially 34 amino acid residues, is highly conserved among ERM proteins and binds to the actin cytoskeleton. A coiled-coil structure connects the N-terminal FERM domain and the C-terminal actin-binding domain. The primary protein structure of EZR and its inactive and active conformations are illustrated in Figure 2.

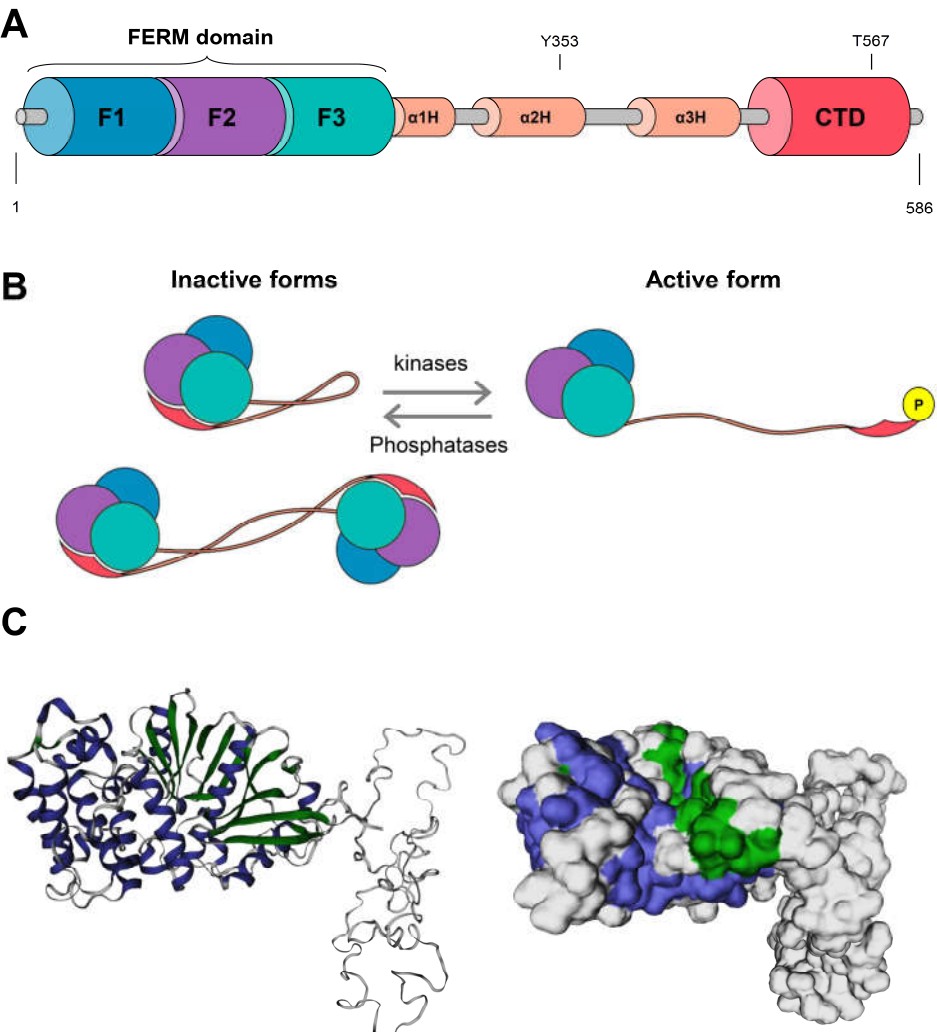

**Figure 2.** Protein structure and activation model for ezrin (EZR). (**A**) The N-terminus consists of a domain called FERM (band 4.1 protein, ezrin, radixin, moesin) formed by three subunits F1, F2, and F3, after which there is a helical domain called α1H, α3H, and α3H followed by a C-terminal domain (CTD) subunit, where the ezrin phosphorylation sites (Y353 and T567) are found. (**B**) Representation of active and inactive forms of EZR. It can interact intermolecularly with itself when inactive, forming a monomer or even a dimer with another protein. When it experiences the action of kinases and is phosphorylated, ezrin is activated, "extends", and passes to a new conformation. The action of phosphatases reverses this conformation. This illustration was adapted from Michie et al. [11]. The three subunits F1 (blue lobe), F2 (purple lobe), and F3 (green lobe) for the FERM domain, helical domain (orange line), and C-terminal domain (pink) are illustrated. (**C**) The 3D reconstitution of the EZR protein was constructed using the SWISS-MODEL platform (https://swissmodel.expasy.org/, accessed on 2 August 2021), and cartoon and surface versions of the protein are illustrated. Alpha-helix structures are shown in purple, beta-pleated sheet structures are in green, and other structures are in white.

## 2. EZR Activation

In the closed inactive state, the FERM domain is tightly associated with the ~80 residues of the C-terminal domain (CTD) from EZR, hiding the membrane association and F-actin-binding sites, and the change from closed to open ERMs requires phosphorylation of a specific threonine residue (T567 in ezrin, T564 in radixin, and T558 in moesin) [12].

Activation of EZR has been proposed to follow a two-step mechanism. The first activation occurs via phosphatidylinositol-4,5-bisphosphate (PIP2) binding at the membrane, which seems to facilitate the binding of EZR to membrane proteins. In other words, PIP2 may activate the dormant protein and expose the membrane binding site. Through this PIP2 binding, T567 in the CTD-ERM association domain becomes accessible for phosphorylation by ROCK, LOK, SLK, and some PKC isoforms [13–16].

### 3. Genomics of EZR

The entire *EZR* gene is approximately 53.6 Kb (start: 158765741 and end: 158819368bp; orientation: reverse strand). In the NCBI database (https://www.ncbi.nlm.nih.gov/gene, accessed on 2 August 2021), there are two transcript variants for *EZR* that encode for the same protein (586 aa). Transcript variant 1 (transcript length: 3069 bp) represents the longer transcript, while transcript variant 2 (transcript length: 3052 bp) differs from transcript variant 1 in the 5′ UTR region. In the Ensembl database (http://www.ensembl.org/, accessed on 2 August 2021), there is one additional transcript variant for EZR (ENST00000392177.8: 2933 bp), which also encodes for the same protein (586 aa).

Recurrent mutations in the *EZR* gene are rare, and most are variants of uncertain significance. Using The Cancer Genome Atlas (TCGA) cohorts (10,953 patients in 32 studies), a total of 192 patients (2%) presented EZR genetic alterations (mutations, amplifications, deep deletions, and multiple alterations), as reported on cBioPortal (http://www. cbioportal.org, accessed on 10 October 2022) (Figure 3A). A total of 110 somatic mutations were found: 84 missense substitutions, 16 truncating, one splice mutation, and nine gene fusions (Figure 3B). Among these mutations, two alterations stand out for having driver mutation potential: *EZR::ROS1* and *ARID1B::EZR* gene fusions.

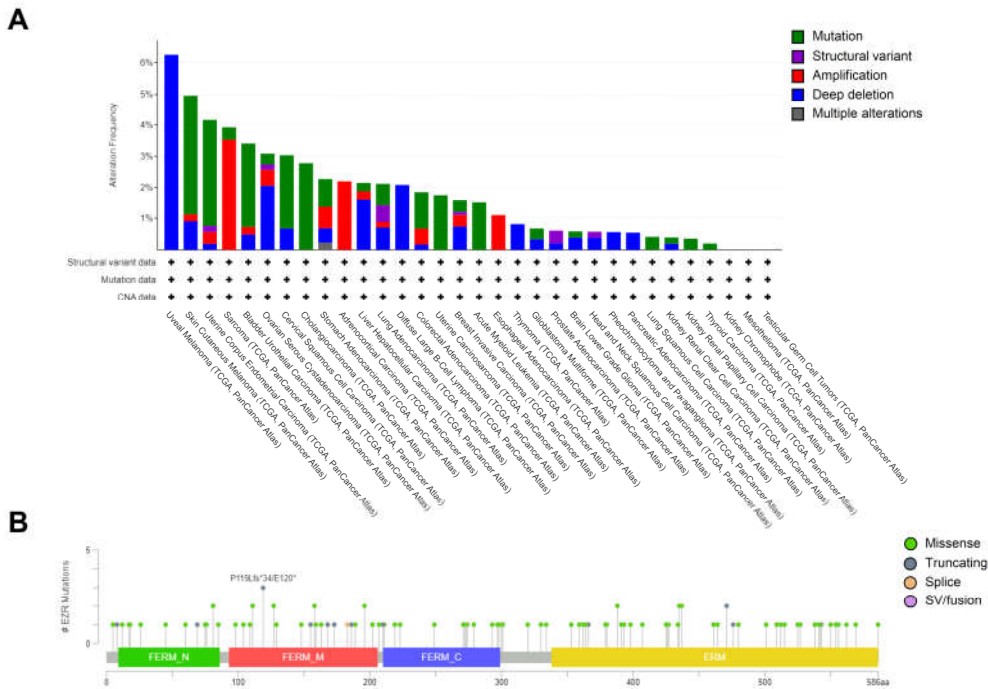

**Figure 3.** *EZR* genomic alterations in cancer. (**A**) Distribution of *EZR* genetic alterations in The Cancer Genome Atlas (TCGA) cohorts (10953 patients in 32 studies). (**B**) A total of 110 somatic mutations were found: 84 missense substitutions, 16 truncating, one splice mutation, and nine gene fusions. The figures were obtained using the cBioPortal (http://www.cbioportal.org, accessed on 2 August 2021).

## 4. Expression and Functions for EZR in Human Cancers

### 4.1. Breast Cancer

EZR is upregulated in breast cancer tissues compared with normal breast tissues, especially in metastatic breast cancer, and also may be used as a biomarker in overall survival (OS) [17]. Indeed, it had been demonstrated that EZR was required for the invasion and metastasis of mammary carcinoma cells [18]. In metastatic breast cancer cells, MDA-MB-231, EZR silencing by shRNA reduced cell motility and invasion, c-Src signaling, and increased E-cadherin expression, a vital component of the adherent junctions [19]. EZR interacts with AKT in breast cancer cells, induces its downstream signaling pathway, and promotes tumor progression [9]. PI3K and proto-oncogene c-Src activities were modulated by EZR and are required for EZR-dependent cell invasion [18]. The co-expression of EZR with CD44, a glycoprotein involved in cell–cell interactions, adhesion, and migration, may also be a potential biomarker for the initiation, progression, and differentiation of breast cancer tumors [20].

### 4.2. Melanoma

Ilmonen et al. [21] showed that EZR expression was found in 76 out of 95 melanoma samples analyzed (80%), with 48 weakly positive and 28 strongly positive, as evidenced by an immunohistochemistry assay. Notably, EZR immunoreactivity of metastatic tumors was more substantial than in primary tumors, and none of the metastasis samples were negative for EZR expression (two samples were weakly positive, and ten were strongly positive). Additionally, EZR expression was associated with tumor proliferation index and tumor growth in primary melanoma [21]. Federici et al. [22] aimed to correlate EZR′s molecular interactions with major actors of the metastatic behavior of tumor cells (i.e., CD44) and showed that an EZR deletion mutant (146 N-terminal amino acids) abolished the in vivo metastatic dissemination of a melanoma model. In melanoma cells, EZR expression was also associated with phagocytic machinery and phagocytosis through actin cytoskeleton modulation [23]. Interestingly, miR-183 may serve as a tumor suppressor since it reduces migratory activity and invasiveness of A375 human melanoma cells. In this context, EZR was identified as a target for miR-183 [24].

### 4.3. Cervical Carcinoma

EZR protein was highly expressed in all cervical cancer cell lines and tissues compared with the normal cervical tissues and predicted poor prognosis in cervical cancer [25,26]. In HPV-associated squamous intraepithelial lesions, EZR expression was associated with the severity of the disease. Intense staining for EZR with diffuse intracellular distribution was also observed in *in situ* adenocarcinoma. In contrast, the distribution in normal endocervical tissue was strongly polarized to the apical side of columnar epithelial cells [27]. In a previous study, the authors observed slightly reduced EZR expression due to HPV 16 E5, an oncogene facilitating the early events in the neoplastic development, suggesting that increased expression of EZR would rather be a consequence of E6 and/or E7 oncogene expression [28]. In HeLa and SiHa cells, EZR silencing by siRNA reduced colony formation and cervical cancer cells′ ability to cross the chorioallantoic membrane surface and infiltrate the underlying stroma, consequently attenuating migration, invasion, mesenchymal marker expression, and PI3K/AKT activation [25].

### 4.4. Colorectal Cancer

Through tissue microarray and immunohistochemistry, Patara et al. [29] concluded that a higher cytoplasmatic EZR expression correlates with higher tumor aggressiveness and poor prognosis. Moreover, its expression is an important prognostic factor in colorectal cancer patients. One of the major causes of poor prognosis in colorectal cancer is lymph node metastasis initiated by epithelial–mesenchymal transition (EMT) due to the

activation of several signaling pathways by EZR, which plays a significant role in protein signal transmission. Indeed, many growth factors and cytokines have been proven essential in stimulating EMT, including EGF. Li et al. [30] showed that the knockout of EZR inhibits EGF-induced lung metastasis of colorectal cancer xenografts. Abnormal activation of EZR and NFκB are related to colorectal cancer metastasis and poor prognosis. EZR expression was also associated with the degree of tumor differentiation, lymph node metastasis, and Dukes′ stage [29,31]. Gavert et al. [32] reported that the NFκB-mediated signaling participates in cellular changes induced by L1 (immunoglobulin-like cell-adhesion receptors) that lead to invasive phenotype in colorectal cancer. Moreover, the same authors found that NFκB- and EZR-mediated signaling are essential for the ability of L1 to induce metastasis in colorectal cancer cells. The decrease in NFκB transactivation, EZR levels, or an L1 mutant with an altered EZR-binding domain blocked the ability of L1 to induce liver metastasis. Accordingly, colorectal cancer patients with moderate to robust EZR expression (by immunohistochemistry) presented a reduced five-year overall survival rate compared to patients with a negative or low-intensity EZR expression [29]. Increased EZR phosphorylation at T567 was also reported in samples from colorectal cancer patients, which was associated with the IGF1R signaling pathway and expression of XIAP and BIRC5, both apoptosis inhibitors [33].

### 4.5. Endometrial Cancer

In endometrial cancer, EZR expression is higher in the high-metastasis Ishikawa subclone (named mEIIL) compared to its parental low-metastasis cell line. Treatment with EZR antisense phosphorothioate oligonucleotides reduced invasiveness, but not proliferation, in mEIIL cells [34]. High EZR protein expression was reported in uterine endometrioid adenocarcinomas. EZR protein levels were also significantly increased in endometrial hyperplasias that more frequently progressed to invasive cancer and metastatic lesions than in the matched primary lesions. In endometrioid carcinomas, EZR was at least focally expressed in 93% of cases, which was quite heterogeneous, with a wide range from only a few to nearly 100% positive tumor cells [35].

### 4.6. Gastric Cancer

EZR immunostaining was positive in 81.1% of intestinal-type and 40.9% of diffuse-type adenocarcinoma cases, suggesting that low EZR expression indicates the loss of adhesion in diffuse carcinomas. Furthermore, EZR overexpression was related to *Helicobacter pylori* infection [36]. In another study, EZR expression was detected, at low levels, in 11.2% of non-tumor mucosae and, at elevated levels, in 59.2% of gastric cancer samples. EZR overexpression was associated with age, tumor size, location, differentiation stage, depth of invasion, vessel invasion, lymph node and distant metastasis, and TNM stage, and it was an independent prognostic factor in patients with gastric carcinoma [37]. Jin et al. reported similar findings [38]. The high frequency of EZR expression suggests a central role in gastric cancer biology. Thus, EZR protein expression could be used as an early diagnostic marker of gastric cancer and its precancerous disease, and its overexpression may be a predictor of poor prognosis [38].

### 4.7. Head and Neck Squamous Cell Carcinoma

In head and neck squamous cell carcinoma patients, high cytoplasmic EZR expression was observed in 92% of the tumors and independently associated with a worse prognosis [39]. These data were validated in a large cohort in which the high EZR expression was associated with poor survival outcomes. Notably, tumors with high cytoplasmic EZR display a more invasive phenotype [40].

### 4.8. Hepatocellular Carcinoma

EZR expression was higher in tumor samples than hepatic tissues and associated with metastasis and differentiation degree [41,42]. High EZR levels have been significantly associated with advanced TNM stage, poor Edmondson's histological grade, macroscopic portal vein invasion, tumor recurrence, and extrahepatic recurrence, and are independently associated with poor overall survival [43,44]. Increased levels of EZR were observed in hepatocarcinoma cell lines with higher metastatic potential, and EZR inhibition by siRNA reduced clonogenicity, cell proliferation, motility, and invasion [44,45]. EZR hyperphosphorylation has been identified as responsible for the invasion of hepatocellular cells, promoting intrahepatic metastasis in vivo and cell migration in vitro [46,47]. EZR overexpression promotes hepatocellular carcinoma cell proliferation, EMT, angiogenesis, and glycolysis reprogramming [46,47]. In hepatocellular carcinoma patients, EZR gene expression was significantly decreased after treatment with arsenic trioxide [48], suggesting a potential approach for pharmacological intervention on EZR expression.

### 4.9. Kidney Cancer

In renal cell cancer, the absence of EZR expression was an independent prognostic factor of disease-specific survival. EZR expression was also associated with incidental tumors, clinical stage, pT stage, synchronic metastasis, and ISUP histological grade [49,50].

### 4.10. Leukemia

In T-cell acute lymphoblastic leukemia cells, ERM phosphorylation has been found to promote invasion [51], and EZR deregulation has been related to the progression of mouse preleukemia erythroblasts toward malignancy [52]. In acute myeloid leukemia cells, functional studies have shown that EZR promotes survival and acts as an effector protein in cell signaling mediated by FLT3-ITD and mutated KIT receptors [52–54]. In a comprehensive analysis of cytoskeleton regulatory genes, Lipreri da Silva et al. [55] identified that high EZR expression is an independent marker of worse outcomes in acute myeloid leukemia patients, and EZR pharmacological inhibition reduced viability, proliferation, autonomous clonal growth, and cell cycle progression in leukemia cells. In chronic lymphocytic leukemia (CLL), EZR is highly expressed and associated with molecular signatures relevant to the disease's development and maintenance, including TP53, PI3K/AKT/mTOR, NFκB, and MAPK pathways [56]. Pharmacological EZR inhibition with NSC305787 reduced viability, clonogenicity, and cell cycle progression and induced apoptosis in CLL primary and cell lines [56].

### 4.11. Lymphoma

In diffuse large B-cell lymphoma, EZR is essential to B-cell antigen receptor organization at the cell membrane and supports proliferation and survival signals. EZR inhibition by siRNA or pharmacological agents reduced cell viability in lymphoma cells [57]. Moreover, it has been shown that ERM proteins regulate B- and T-cell activation by controlling BCR and TCR dynamics, scaffolding protein assembly, and membrane-associated intracellular signaling [58].

### 4.12. Lung Cancer

EZR expression was associated with late-stage tumors, pleural invasion, and poor survival outcomes in non-small cell lung carcinomas [59]. EZR phosphorylation at T567 and Y353 was significantly upregulated in non-small cell lung carcinomas compared with normal tissues, which was also correlated with poor differentiation and late clinical stage. However, only EZR phosphorylation at T567 overexpression was associated with the presence of lymph node metastasis and overall survival [60]. EZR may also be a possible

predictor to assess the prognosis in patients with non-small cell lung carcinoma since its gene expression was associated with lymphatic metastasis [61].

### 4.13. Malignant Pleural Mesothelioma

Pleural mesothelioma expresses activated ERM proteins, with EZR being the principal component of this family of proteins associated with proliferation and migration in this disease [62].

### 4.14. Oral Squamous Cell Carcinoma

EZR expression was significantly associated with N-classification, UICC stage, and lymphangitic carcinomatosis [63]. Another study, using tissue microarray, observed that EZR is overexpressed in primary tongue squamous cell carcinoma and may be implicated in these cells′ growth, migration, and invasiveness, possibly regulating the E-cadherin/β-catenin complex [64]. Evaluation of EZR expression in human tongue cancer tissues by immunohistochemical staining showed that this protein was highly expressed in invasive squamous cell carcinoma compared to in situ carcinoma [65]. Another study also suggested that EZR may be involved in the progression of tongue carcinoma in situ to squamous cell carcinoma [66]. Immunoexpression results confirmed a correlation between EZR expression in squamous cell carcinoma of the lip, suggesting cooperative participation of these proteins in cell movement and invasion [67]. In tongue squamous cell carcinoma clinical samples, EZR activation at Y353 is associated with metastases and poor patient prognosis [68].

### 4.15. Ovarian Cancer

In OVCA cells, estradiol increased EZR expression, which was associated with invasive behavior [69]. In ovarian carcinoma patients, EZR expression presents a high predictive value for tumor-related death, including late stage [70]. EZR regulates OVCA cell proliferation, migration, and invasion by modulating EMT and inducing the formation of actin stress fibers by regulating Rho GTPase activity [71]. In three-dimensional cultured ovarian carcinoma cell lines, reduced expression of EZR by shRNA impaired invasiveness and cell branching ability [72].

### 4.16. Pancreatic Cancer

Zhou et al. [73] showed that the levels of p-EZR T567/p-EZR Y353 protein expression in the cytoplasm of pancreatic cancer cells increased with the TNM stage of human pancreatic cancers. The survival time of the group positive for p-EZR T567/p-EZR Y353 protein expression was shorter than that of the negative group [73]. EZR levels were significantly upregulated in pancreatic cancer patients′ samples and correlated with poor prognosis [74]. Two active phosphorylated sites of EZR, Y353 and T567, were differentially expressed in pancreatic cancer tissues. High EZR phosphorylation at Y353, but not T567, was associated with malignant progression and clinical outcomes of pancreatic cancer patients [75]. In pancreatic cancer cells, EZR was involved in the cytoskeleton modulation, protrusions, and microvilli, which were related to migration and invasion [76]. Bioinformatic tools showed that EZR expression predicted worse survival outcomes in the pancreatic adenocarcinoma cohort, and gene signatures were associated with EZR expression, including apoptosis, PI3K/AKT/mTOR signaling, estrogen response early, NOTCH signaling, estrogen response late, and pancreatic beta cells [77]. Pharmacological EZR inhibitor, NSC305787, reduced viability, clonogenicity, migration, and induced apoptosis in pancreatic cancer cells [77].

### 4.17. Prostate Cancer

EZR expression was moderate or strong in samples from prostate cancer patients and negative or weakly expressed in benign cases [78]. EZR staining was more intense in high-grade prostatic intraepithelial neoplasia than in other prostate cancer cells [79]. EZR was found to be an essential intracellular mediator of cell invasion in hormone-sensitive and hormone-resistant prostate cancer cells. Androgen treatment induces EZR expression and phosphorylation at T567 and Y353 residues. Overexpression of mutated EZR variants (T567A and Y353F) or EZR silencing by siRNA inhibited androgen-induced invasion [80]. In metastatic prostate cancer cell lines (22RV1 and PC-3), EZR overexpression promoted migratory and invasive abilities. Accordingly, EZR expression was significantly increased in circulating tumor cells from metastatic prostate cancer patients [81].

After radical prostatectomy, circulating tumor cells from EZR-positive prostate cancer patients were susceptible to tumor metastases [81]. In nude mice bearing prostate cancer, treatment with baicalein decreased tumor volume and weight, which were much more pronounced in those with in vivo EZR knockdown [82]. Furthermore, baicalein treatment suppresses proliferative capacity in prostate cancer cell lines, interrupts the cell cycle progression, and stimulates apoptosis through EZR downregulation [82].

## 5. Concluding Remarks and Pharmacological Advances for EZR Inhibition

In most tumors studied to date, EZR is highly expressed and predicts an unfavorable clinical outcome. Studies with genetic intervention in experimental models highlight the importance of EZR in maintaining the malignant phenotype and directing this gene/protein (or the signaling mediated by it) as a potential target for antineoplastic therapy.

Two synthetic compounds that act as selective EZR inhibitors (NSC668394 and NSC305787) have been evaluated in preclinical studies (Figure 4). These compounds directly bind to EZR, inhibiting T567 phosphorylation with low micromolar affinity [83]. It has also been demonstrated that these EZR inhibitors have distinct target-binding profiles. For example, NSC305787 bound and dissociated more quickly from EZR than NSC668394 [83]. Another point worth mentioning is that NSC305787 and NSC668394 are well tolerated in murine models without any apparent acute toxicity [83]. Regarding pharmacokinetics, NSC305787 has the most prolonged plasma half-life (>6 h by intraperitoneal (i.p.) injection) compared to NSC668394 (<0.5 h i.p.) [84]. Although these compounds have potent antineoplastic activity in models of osteosarcomas, lung cancer, leukemia, and pancreatic cancer, their underlying molecular mechanisms and potential application in other cancer models remain poorly explored [55,77,83].

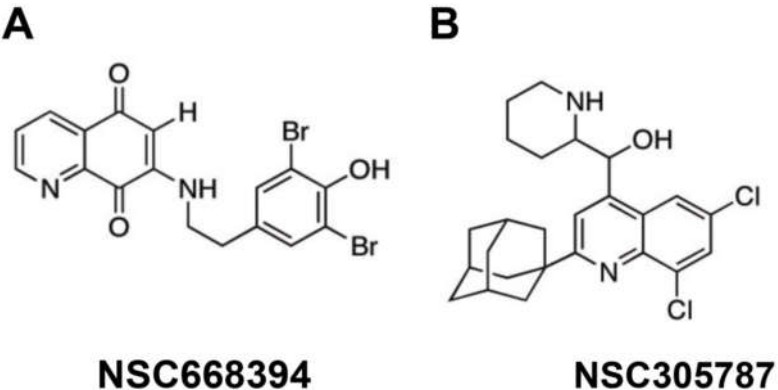

**Figure 4.** Chemical structures of putative pharmacological EZR inhibitors NSC668394 (**A**) and NSC305787 (**B**).

A summary of EZR expression in cancer is presented in Table 1, and a summary of EZR functional assays in cancer cells is shown in Table 2.

**Table 1.** Summary of EZR expression in cancer.

| Publications | Cancer Type | Cell Lines | Primary Tissues | Techniques | Notes |
|---|---|---|---|---|---|
| Zhang et al. [17] | Breast cancer | No | Yes, 120 samples from breast cancer patients. | IHC | EZR was upregulated in breast cancer and may be used as a potential biomarker for overall survival. |
| Li et al. [19] | Breast cancer | Yes, MCF-7, MDA-MB-453, MDA-MB-435s, and MDA-MB-231 | Yes, tumor samples from 23 patients with invasive human ductal breast cancer. | RT-PCR, IHC | EZR protein expression was significantly higher in primary cancer tissues than in lymph node metastases. |
| Ilmonen et al. [21] | Melanoma | No | Yes, 95 samples from patients with primary melanoma of the skin. | IHC | EZR expression correlated with tumor thickness and level of invasion. |
| Kong et al. [26] | Cervical cancer | No | Yes, 586 samples were collected from routinely processed and diagnosed uterine cervical lesion tissues from pretreatment surgical resections. | qRT-PCR, IHC | EZR was a potentially effective predictor of the poor prognosis of cervical cancer patients, especially those with early-stage disease. The determination of EZR expression levels may help to identify high-risk cervical cancer. |
| Patara et al. [29] | Colorectal cancer | No | Yes, samples from tumors and construction of a TMA. | TMA, IHC | Increased cytoplasmatic EZR expression correlated with higher tumor aggressiveness and a worse prognosis in colorectal cancer patients. |
| Gavert et al. [32] | Colorectal cancer | Yes, Ls174T and SW620 | Yes, samples from tumor and normal colon tissues. | | Invading tumor cells expressed high EZR levels. |
| Leiphrakpam et al. [33]. | Colorectal cancer | Yes, GEO, GEORI, CBS, HCT166, and HCT166b | Yes, samples of tumors and construction of a TMA. | TMA, IHC | Increased expression of p-EZR[T567] was found in liver metastasis of orthotopic implantation studies in vivo and IHC studies of human CRC patient specimens. |
| Ohtani et al. [85] | Endometrial cancer | No | Yes, 20 samples from uterine endometrioid adenocarcinoma, 7 samples from simple endometrial hyperplasias, 7 samples from complex endometrial hyperplasias, 7 samples from | IHC | EZR protein was explicitly expressed in uterine endometrioid adenocarcinoma and its precursor lesions. EZR protein expression in metastatic lesions increased and shifted from the cytosol to the membrane. |

| | | | | | |
|---|---|---|---|---|---|
| | | | atypical endometrial hyperplasias, and 12 samples of normal endometrium. | | |
| Bal et al. [36]. | Gastric cancer | No | Yes, samples from 53 intestinal-type adenocarcinoma and 22 diffuse-type carcinoma patients. | IHC | *H. pylori*-infected gastric carcinomas presented increased EZR expression. |
| Li et al. [37] | Gastric cancer | No | Yes, samples from 436 gastric cancer patients. | TMA, IHC | EZR was upregulated in gastric cancer tissues compared with normal gastric tissues and correlated significantly with prognosis. In addition, high levels of EZR expression were associated with age, tumor size, location, depth of invasion, vessel invasion, lymph node and distant metastasis, and TNM stage. |
| Schlecht et al. [40] | Head and neck squamous cell carcinoma | No | Yes, samples from 128 primary HNSCC were collected at initial diagnosis and treatment. | TMA, global RNA expression | No association with prognosis for HNSCC was found due to the inherent heterogeneity in disease management. |
| Lee et al. [59] | Lung cancer | No | Yes, 112 NSCLC specimens. | TMA, IHC | EZR was overexpressed in tumor tissues. EZR expression correlated with pleural invasion and pathological stage. The negative group for EZR expression presented more prolonged survival. |
| Jin et al. [60] | Lung cancer | No | Yes, samples from 150 NSCLC tumors and 14 normal lungs. | IHC, qRT-PCR | EZR was upregulated in NSCLC. Increased EZR levels correlated with NSCLC late stage and poor differentiation. p-EZR$^{T567}$ correlated with the presence of lymph node metastases. EZR was associated with reduced survival time for patients with early-stage NSCLC. |
| Kolegova et al. [61] | Lung cancer | No | Yes, samples from 46 NSCLC patients | qRT-PCR, Western blotting | Increased EZR gene and protein expression was found in patients with distant metastasis. |
| Lipreri da Silva et al. [55] | Acute myeloid leukemia | Yes, Kasumi-1 and MOLM-13 | Yes, the cohort deposited in TCGA. | Bioinformatics | EZR expression was a prognostic marker in AML. In intermediate-risk AML patients, high EZR expression distinguished a group with a poor prognosis. |

| | | | | | |
|---|---|---|---|---|---|
| Lipreri da Silva et al. [56] | Chronic lymphocytic leukemia | Yes, MEC-1 | Yes, samples from 56 CLL patients and ten age-matched healthy donors. | Bioinformatics, cDNA microarray, qPCR | EZR was highly expressed and positively associated with relevant signaling pathways related to CLL development and progression, including TP53, PI3K/AKT/mTOR, NFκB, and MAPK. |
| Safi et al. [63] | Oral squamous cell carcinoma | No | Yes, resection specimens from 80 treatment-naive OSCC patients. | IHC | EZR expression had a significant impact on overall survival. Increased EZR expression was associated with malignant progression, leading to a higher risk of metastases. |
| Saito et al. [64] | Oral squamous cell carcinoma | Yes, HSC-3 | Yes, 79 samples (10 normal tongue tissues and 69 tongue SCC tissues) | IHC, microarray | EZR was overexpressed in 46.4% of the tumors examined. EZR expression was correlated with proliferative activity. |
| Noi et al. [65] | Oral squamous cell carcinoma | Yes, HSC-3 and HSC-4 (3D culture) | Yes, human tongue cancer tissue (CIS (12 cases) and invasive SCC of the tongue (T1: ten cases, T2: three cases)). | IHC | Higher expression of EZR was found in invasive squamous cell carcinoma than in carcinoma in situ. |
| Noi et al. [66] | Oral squamous cell carcinoma | No | Yes, in situ tongue carcinoma patients (CIS, n = 17) and tongue squamous cell carcinoma patients (SCC, n = 46). | IHC | EZR expression was highly expressed in the cell membrane. EZR appears to be involved in the progress from in situ carcinoma in the tongue into squamous cell carcinoma. |
| Assao et al. [58] | Oral squamous cell carcinoma | No | Yes, samples of 91 patients with primary squamous cell carcinoma of the lower lip. | IHC | EZR expression in squamous cell carcinoma of the lip suggests the participation of this protein in cell movement and invasion. |
| Wang et al. [68] | Oral squamous cell carcinoma | Yes, SCC9 and SCC25 | Yes, primary tongue carcinomas collected from 63 patients. | IHC | EZR may be a therapeutic target to reverse EMT in tongue cancers and prevent TSCC progression. EZR activation was associated with metastasis and poor patient survival in TSCCs. |
| Köbel et al. [70] | Ovarian cancer | Yes, SKOV-3 | Yes, ovarian carcinoma samples from 105 patients. | IHC | EZR was overexpressed in 49% of the samples. EZR expression correlated with reduced overall survival and appeared as an independent risk factor. |
| Li et al. [71] | Ovarian cancer | Yes, SKOV3 and CaOV3 | No. | Western blotting, qPCR | EZR was overexpressed in either CaOV3 or SKOV3. |

| Horwitz et al. [72]. | Ovarian cancer | Yes, ES2 and OVCAR3 | Yes, 93 effusions (76 peritoneal, 17 pleural) from 93 patients diagnosed with high-grade serous carcinoma (HGSC). | qRT-PCR | High expression of EZR was found in tumors. EZR expression in effusions was unrelated to clinical outcome. |
|---|---|---|---|---|---|
| Zhou et al. [73] | Pancreatic cancer | Yes, SW1990 | Yes, 19 samples were obtained from patients undergoing surgical resection. | IHC, plasmid transfection, target gene expression | The expression of phosphorylated EZR proteins was related to pancreatic cancer's clinical and pathological features. Phosphorylated EZR induced a positive regulatory role in growth, adhesion, and invasion. |
| Liprei da Silva et al. [77] | Pancreatic cancer | No | Yes, the cohort was deposited in the TCGA. | Bioinformatics | High EZR expression predicted worse survival outcomes in the pancreatic adenocarcinoma cohort. |
| Valdman et al. [78] | Prostate cancer | No | Yes, 103 radical prostatectomy specimens. | IHC | EZR immunoreactivity in prostate cancers was moderate or strong in 70% of specimens. EZR expression was correlated with Gleason score and seminal vesicle invasion. |
| Pang et al. [79] | Prostate cancer | No | Yes, 19 high-grade prostatic intraepithelial neoplasia (HGPIN) samples obtained from radical prostatectomy specimens. | IHC | Immunoreactivity for EZR was absent or weak in benign prostatic epithelial cells. The immunostaining was moderate or strong in all HGPIN samples. |
| Chen et al. [81] | Prostate cancer | Yes, 22RV1 and PC-3 | Yes, samples from 80 prostate cancer patients. | qRT-PCR and IHC | Ezrin was significantly increased in prostate cancer tissues and 22RV1 and PC-3 cell samples. EZR expression in CTCs from patients with prostate cancer was significantly increased with metastatic grade. |

Abbreviations: EZR, ezrin; IHC, immunohistochemistry; RT-PCR, reverse transcription polymerase chain reaction; qRT-PCR, quantitative reverse transcription polymerase chain reaction; TMA, tissue microarray; TNM, TNM classification of malignant tumors; HNSCC, head and neck squamous cell carcinoma; NSCLC, non-small cell lung carcinoma; AML, acute myeloid leukemia; CLL, chronic lymphocytic leukemia; OSCC, oral squamous cell carcinoma; TSCC, tongue squamous cell carcinoma; HGSC, high-grade serous carcinoma; CTC, circulating tumor cells.

**Table 2.** Summary of EZR functional assays in cancer cells.

| Publication | Cancer Type | Cell Lines | Approach | Activity | Notes |
|---|---|---|---|---|---|
| Li et al. [19] | Breast cancer | MCF-7, MDA-MB-453, MDA-MB- | shRNA | EZR downregulation | Decreased EZR expression reversed metastatic behaviors of human breast cancer cells by |

| | | | | | |
|---|---|---|---|---|---|
| | | 435s, and MDA-MB-231. | | | inducing c-SRC-mediated E-cadherin expression. |
| Federici et al. [22] | Melanoma | MM1, MM2, MM3, and HeLa. | Stable transfection | EZR deletion mutant | Expression of EZR deletion mutant comprising 146 N-terminal amino acids abolished metastatic dissemination. |
| Zhang et al. [24] | Melanoma | A375. | MiR-183 overexpression and knockdown | EZR downregulation | miR-183 inhibits A375 human melanoma cell migration and invasion, possibly through the downregulation of EZR. |
| Kong et al. [25] | Cervical carcinoma | HeLa, SiHa, C33A, and CaSki. | RNAi | EZR downregulation | EZR silencing inhibited the proliferation, migration, and invasion of uterine cervical cancer cells through epithelial–mesenchymal transition inhibition. |
| Li et al. [30] | Colorectal cancer | SW480 and SW116. | siRNA and shRNA | EZR downregulation | Inhibition of EZR reduced EGF-induced epithelial–mesenchymal transition and lung metastasis of colorectal cancer. |
| Leiphrakpam et al. [3] | Colorectal cancer | GEO, GEORI, CBS, HCT166, and HCT166b. | NSC668394 and NSC305787 | Pharmacological inhibition of EZR phosphorylation | EZR inhibitors were effective in dephosphorylating EZR at T567 and decreased XIAP levels in metastatic colorectal cancer cells. |
| Ohtani et al. [34]. | Endometrial cancer | Ishikawa and mEIIL. | ePONs | EZR downregulation | Inhibitory effect of ePONs on invasiveness results from EZR suppression. |
| Lipreri da Silva et al. [55] | Acute myeloid leukemia | Kasumi-1 and MOLM-13. | NSC305787 and NSC668394 | Pharmacological inhibition of EZR phosphorylation | EZR inhibition reduced cell viability, clonogenicity, phosphorylation of the AKT signaling pathway and increased apoptosis in acute myeloid leukemia cell lines. |
| Lipreri da Silva et al. [56] | Chronic lymphocytic leukemia | MEC-1 and primary patients' cells. | NSC305787 | Pharmacological inhibition of EZR phosphorylation | NSC305787 reduced viability, clonogenicity, and cell cycle progression and induced apoptosis in chronic lymphocytic leukemia cells. Pharmacological EZR inhibition also attenuated ERK, S6RP, and NFκB activation. |
| Saito et al. [64] | Oral squamous cell carcinoma | HSC-3. | RNAi | EZR downregulation | EZR depletion significantly reduced cell proliferation, migration, and invasiveness and disturbed actin reorganization during podia formation. |
| Noi et al. [5] | Oral squamous cell carcinoma | Yes, HSC-3 and HSC-4 (3D culture). | siRNA | EZR downregulation | No marked morphological differences were observed upon EZR silencing. |

| | | | | | |
|---|---|---|---|---|---|
| Wang et al. [68]. | Oral squamous cell carcinoma | SCC9 and SCC25. | siRNA and lentivirus-mediated shRNA | EZR downregulation | EZR silencing reduced the invasion and migration of SCC9 and SCC25 cells. Downregulation of EZR also inhibited EGF-induced EMT in tongue squamous carcinoma cells. |
| Song et al. [69] | Ovarian cancer | SKOV3 and DOV13. | Estradiol treatment | Estrogen-induced EZR overexpression | Estrogen induced EZR over-expression and the invasiveness of OVCA cells in culture. |
| Köbel et al. [70] | Ovarian cancer | SKOV-3. | siRNA | EZR downregulation | EZR silencing reduced cell invasion in vitro. |
| Li et al. [71] | Ovarian cancer | SKOV3 and CaOV3. | siRNA and FLAG-EZR overexpression plasmid | EZR downregulation and EZR overexpression | EZR ectopic expression increased cell proliferation, invasiveness, and epithelial–mesenchymal transition. EZR knockdown prevented cell proliferation, invasiveness, and epithelial–mesenchymal transition. |
| Horwitz et al. [72]. | Ovarian cancer | ES2 and OVCAR3. | Lentivirus-mediated shRNA | EZR downregulation | Reduced EZR expression impaired ovarian cancer cells' invasion ability and branching capacity. |
| Zhou et al. [73]. | Pancreatic cancer | SW1990. | Recombinant plasmids | EZR overexpression | Overexpression of T567D ezrin (p-EZR), a mutant that mimics permanent phosphorylation, in SW1990 cells increased proliferative capacity and growth rate. |
| Liprei da Silva et al. [77] | Pancreatic cancer | PANC-1, AsPC-1, and MIA PaCa-2. | NSC305787 | Pharmacological inhibition of EZR phosphorylation | Inhibition of EZR favors a molecular network that reduces proliferation and induces apoptosis in pancreatic cancer cells. |
| Chuan et al. [80] | Prostate cancer | LNCaP-FGC, PC-3 and PCa. | Androgen treatment, siRNA and VSV-G-tagged human wild-type EZR and Y353F-EZR mutant | Androgen-induced EZR expression, EZR downregulation, and EZR overexpression | Androgen treatment induces EZR expression and phosphorylation of ezrin in T567 and Y353. EZR inhibition function using short interference RNA or the overexpression of T567A and Y353F-EZR mutants significantly reduces androgen-induced Matrigel invasion. Androgens regulate EZR at transcriptional and posttranscriptional levels. |
| Chen et al. [81] | Prostate cancer | 22RV1 and PC-3. | EZR overexpression and si-EZR plasmids | EZR downregulation and EZR overexpression | EZR overexpression promoted the migratory and invasive abilities of 22RV1 and PC-3 cells. |

Abbreviations: EZR, ezrin; RNAi, RNA interference; siRNA, small interfering RNA; shRNA, short hairpin RNA; ePONs, antisense phosphorothioate oligonucleotides.

**Author Contributions:** Conceptualization, J.C.L.d.S. and J.A.M.-N.; writing—original draft preparation, J.C.L.d.S. and J.A.M.-N.; writing—review and editing, H.P.V.; visualization, J.C.L.d.S. and H.P.V.; supervision, J.A.M.-N.; project administration, J.A.M.-N.; funding acquisition, J.A.M.-N. All authors have read and agreed to the published version of the manuscript.

**Funding:** J.C.L.d.S. and H.P.V. received fellowships from the São Paulo Research Foundation (FAPESP; grants #2020/12909-7 and #2021/01460-1). This study was supported by grants #2019/23864-7 and #2021/11606-3 from the FAPESP. This study was financed in part by the Coordenação de Aperfeiçoamento de Pessoal de Nível Superior, Brasil (CAPES), Finance Code 001.

**Institutional Review Board Statement:** Not applicable.

**Informed Consent Statement:** Not applicable.

**Data Availability Statement:** Not applicable.

**Conflicts of Interest:** The authors declare no conflict of interest.

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
