# Peer review of "Perspectives for Targeting Ezrin in Cancer Development and Progression"

_futurepharmacol, doi:10.3390/futurepharmacol3010005_

Round 1

Reviewer 1 Report

Authors reviewed the pathological roles of ezrin (EZR) in cancer development. They also introduced EZR as a potential target molecule for antineoplastic therapy and introduced EZR inhibitors. The review may give readers comprehensive information from the clinical points of view.  However, the molecular mechanism underlying cancer progression by EZR is not well introduced and not satisfactory from the viewpoints of readers. My review comments are shown below.

Major comments:

1.

There is almost no explanation (or not enough) about the signaling pathways involving EZR although authors presents them in the right half of Figure 1.  Authors should explain the signaling pathways (at least in brief) in order that readers can understand the signaling pathways in which EZR is included.  

2.

Authors should mention about the important roles of phosphorylation at Tyr353 (Y353) in addition to those of phosphorylation at Thr567 (T567) in oral squamous cell carcinoma (3.14), pancreatic cancer (3.16), and prostate cancer (3.17). However, they do not introduce the roles and location of this residue in the introduction section. In fact, Y353 is a very important residue for PI3K/AKT/mTOR signaling. Authors should mention about Y353 appropriately. Please check the paper shown below.  

Gautreau A. et al. Proc. Natl. Acad. Sci. USA 1999, 96, 7300–7305.

3.

Authors do not mention about the roles of PIP2 on the activation process of EZR in the introduction section. PIP2 binding is important in the process of EZR activation before phosphorylation. They should mention about this point. 

4.

In Figure 2C, authors should explain the figure. Readers cannot understand this figure at all. Where are F1, F2 and F3 lobes? What do green and blue/violet parts represent?

5.

Author had better add the following paper in the section 4.

Da Silva JCL et al. Life Sci. 2022 Nov 3;311(Pt B):121146.

Minor comments:

In Figure 2B the color of the balls are different between the left and right side of figures. I wonder whether there is any meaning or not.

Author Response

Reviewer #1: Authors reviewed the pathological roles of ezrin (EZR) in cancer development. They also introduced EZR as a potential target molecule for antineoplastic therapy and introduced EZR inhibitors. The review may give readers comprehensive information from the clinical points of view. However, the molecular mechanism underlying cancer progression by EZR is not well introduced and not satisfactory from the viewpoints of readers. My review comments are shown below.

1. There is almost no explanation (or not enough) about the signaling pathways involving EZR although authors presents them in the right half of Figure 1. Authors should explain the signaling pathways (at least in brief) in order that readers can understand the signaling pathways in which EZR is included.

Authors' response: The authors thank the Reviewer for the opportunity to improve this topic. Up to this point, we have added a brief explanation about two major pathways activated by ERM proteins, PI3K/AKT/mTOR implicated in cell proliferation and Rho GTPases involved in cell migration and invasion, as illustrated in Figure 1.

"EZR plays an essential role as an activator of notable signal transduction pathways involved in cancer progression, like PI3K/AKT/mTOR signaling, which provides a mechanism to anchor PI3K in the proximity of its substrate recruiting the p85 regula-tory subunit. In contrast, the phosphorylation of EZR at Y353 is also essential to acti-vate the PI3K signaling [1,2,3,4,5]. The activation of AKT protects cells from apoptosis by phosphorylating and inactivating BAD, a proapoptotic member of the BCL2 family, and increasing cell proliferation [6]. Furthermore, ERM phosphorylation, which ap-pears to be regulated positively by Rho and (possibly) negatively by RAC, may activate downstream signaling of Rho proteins (including RAC) that are required for mem-brane ruffling and lamellipodium extension and CDC42 that induces the formation of filopodia, both GTPases are essential for cell migration and invasion [7,8]

2. Authors should mention about the important roles of phosphorylation at Tyr353 (Y353) in addition to those of phosphorylation at Thr567 (T567) in oral squamous cell carcinoma (3.14), pancreatic cancer (3.16), and prostate cancer (3.17). However, they do not introduce the roles and location of this residue in the introduction section. In fact, Y353 is a very important residue for PI3K/AKT/mTOR signaling. Authors should mention about Y353 appropriately. Please check the paper shown below.

Gautreau A. et al. Proc. Natl. Acad. Sci. USA 1999, 96, 7300–7305.

Authors' response: The authors thank the reviewer for this important observation. We appropriately mentioned that the phosphorylation at Y353 is essential to activate PI3K and, consequently, the downstream signaling.

3. Authors do not mention about the roles of PIP2 on the activation process of EZR in the introduction section. PIP2 binding is important in the process of EZR activation before phosphorylation. They should mention about this point.

Authors' response: Reviewer #2 also suggested a section discussing EZR activation, so the importance of the PIP2 was included in this new section.

"EZR activation

In the closed inactive state, the FERM domain is tightly associated with the ~80 residues of the C-terminal domain (CTD) from EZR, hiding the membrane association and F-actin-binding sites, and the changing from closed to open ERMs requires phosphorylation of a specific threonine residue (T567 in ezrin; T564 in radixin and T558 in moesin) [11]. Activation of EZR has been proposed to follow a two-step mechanism. The first activation occurs via phosphatidylinositol-4,5-bisphosphate (PIP2) binding at the membrane, which seems to facilitate the binding of EZR to membrane proteins. In other words, PIP2 may activate the dormant protein and expose the membrane bind-ing site. Through this PIP2 binding, T567 in the CTD-ERM association domain becomes accessible for phosphorylation by Rho kinase and some PKC isoforms [12,13,14].”

4. In Figure 2C, authors should explain the figure. Readers cannot understand this figure at all. Where are F1, F2 and F3 lobes? What do green and blue/violet parts represent?

Authors' response: Thank you for the opportunity to describe our Figures better. In the revised version of the manuscript, we have included more details about this in the legend.

5. Author had better add the following paper in the section 4.

Da Silva JCL et al. Life Sci. 2022 Nov 3;311(Pt B):121146.

Authors' response: The article's citation is in the "leukemias" section and the tables.

6. In Figure 2B the color of the balls are different between the left and right side of figures. I wonder whether there is any meaning or not.

Authors' response: We have reintroduced the colors in Figure 2B to ensure color consistency. Thank you for the note.

Reviewer 2 Report

This review is well organized, and easy to follow, making it a good resource for scientists new to Ezrin biology.

I suggest some minor revisions:

Figure 1 is less attractive than all the following figures and less informative, The receptors the are involved with Ezrin activation are not labeled or mentioned either in the figure or in the text.

The authors might consider a section discussing Ezrin activation.

The section on colorectal cancer would benefit from including a discussion of findings in the article Gavert et al 2010 JCS.

Also the section discussing inhibitors of Ezrin should include the reference Celik et al 2015 MCT.

The authors should include a section describing any clinical trials using Ezrin inhibitors and make a stronger closing statement about future directions.

Author Response

Reviewer #2: This review is well organized, and easy to follow, making it a good resource for scientists new to Ezrin biology.I suggest some minor revisions:

Authors' note: Thank you very much for the opportunity to improve our manuscript. We added the points proposed in the revised version of the manuscript and hope it is now clearer.

1. Figure 1 is less attractive than all the following figures and less informative, The receptors the are involved with Ezrin activation are not labeled or mentioned either in the figure or in the text.

Authors' response: The primary receptors that act in the EZR signaling and showed in the cell surface in Figure 1 were labeled and mentioned in the Figure legend in the current version of the manuscript.

2. The authors might consider a section discussing Ezrin activation.

Authors' response: It is essential to discuss the activation of EZR and the role of PIP2. A topic discussing EZR activation was included in the revised version of the manuscript.

"EZR activation

In the closed inactive state, the FERM domain is tightly associated with the ~80 residues of the C-terminal domain (CTD) from EZR, hiding the membrane association and F-actin-binding sites, and the changing from closed to open ERMs requires phosphorylation of a specific threonine residue (T567 in ezrin; T564 in radixin and T558 in moesin) [11]. Activation of EZR has been proposed to follow a two-step mechanism. The first activation occurs via phosphatidylinositol-4,5-bisphosphate (PIP2) binding at the membrane, which seems to facilitate the binding of EZR to membrane proteins. In other words, PIP2 may activate the dormant protein and expose the membrane bind-ing site. Through this PIP2 binding, T567 in the CTD-ERM association domain becomes accessible for phosphorylation by Rho kinase and some PKC isoforms [12,13,14].”

3. The section on colorectal cancer would benefit from including a discussion of findings in the article Gavert et al 2010 JCS.

Authors' response: The authors are grateful for the indication of the reference. It is indeed a fascinating article and closely related to our review. We have included the citation of this article in the current version of the manuscript.

" Gavert et al. [30] reported that the NFκB-mediated signaling participates in cellular changes induced by L1 (immunoglobulin-like cell-adhesion receptors) that lead to in-vasive phenotype in colorectal cancer. Moreover, the same authors found that NFκB- and EZR-mediated signaling are essential for the ability of L1 to induce metastasis in colorectal cancer cells. The decrease of NFκB transactivation, EZR levels, or an L1 mu-tant with an altered EZR-binding domain blocked the ability of L1 to induce liver metastasis.

4. Also the section discussing inhibitors of Ezrin should include the reference Celik et al 2015 MCT. The authors should include a section describing any clinical trials using Ezrin inhibitors and make a stronger closing statement about future directions.

Authors' response: The authors thank the reviewer for the opportunity to clarify this topic. Currently, there is no clinical trial using EZR inhibitors. There is a pre-clinical study in osteosarcoma elucidating that the inhibitors did not show toxicity and that NSC305787 has better pharmacokinetics than NSC668394. We believe that the use of these inhibitors is promising in this and other neoplasms. An ex-vivo study carried out by our group also showed that the use of the NSC305787 inhibitor is effective in decreasing viability in primary cells of chronic lymphocytic leukemia.

Round 2

Reviewer 1 Report

The paper is well improved. However, there are still some problems in the revised manuscript. I wish authors to check and brush up their manuscript from the viewpoints of readers. 

Line 48-51

Sentences of Figure 1 legend (especially 3rd sentence from lines 48 to 52) are too long to understand easily. Authors should check and brush up their manuscript. 

Line 72-73

T567 in the CTD-ERM association …..by Rho kinase and some PKC isoforms.

Authors should mention about phosphorylation by LOK/SLK in addition to Rho kinase and PKCs. Viswanatha, R. et al. reported that LOK and SLK are the major kinases involved in ezrin phosphorylation (Viswanatha, R. et al. J. Cell Biol. 2012, 199, 969–984).

Figure 2

Authors should show Y353 in Figure 2A.

I strongly recommend that authors had better to use same colors in Figure 2A and 2B (F1, F2, F3 and CTD).

.  

In Figure 2C, structures shown in white is not clear. Authors should use different color or bold lines.

Author Response

Reviewer #1: The paper is well improved. However, there are still some problems in the revised manuscript. I wish authors to check and brush up their manuscript from the viewpoints of readers.

1. Line 48-51: Sentences of Figure 1 legend (especially 3rd sentence from lines 48 to 52) are too long to understand easily. Authors should check and brush up their manuscript.

Authors’ response: The authors are grateful for the observation. We changed the sentence to improve the concatenation of the information.

2. Line 72-73: T567 in the CTD-ERM association …..by Rho kinase and some PKC isoforms. Authors should mention about phosphorylation by LOK/SLK in addition to Rho kinase and PKCs. Viswanatha, R. et al. reported that LOK and SLK are the major kinases involved in ezrin phosphorylation (Viswanatha, R. et al. J. Cell Biol. 2012, 199, 969–984).

Authors’ response: The authors are immensely grateful to the reviewer for this valuable contribution. We have added these kinases in the current version of the manuscript as well as the suggested reference.

3. Figure 2: Authors should show Y353 in Figure 2A.

Authors’ response: We have added this information in the new version of the Figure.

4. I strongly recommend that authors had better to use same colors in Figure 2A and 2B (F1, F2, F3 and CTD).

Authors’ response: In the current version, the same colors were used in Figure 2A and 2B.

5. In Figure 2C, structures shown in white is not clear. Authors should use different color or bold lines.

Authors’ response: The authors agree with the Reviewer. In this way, we increased the figure's contrast, which greatly improved the visualization of the white parts of the figure.